Linking watershed nutrient loading to estuary water quality with generalized additive models

Schramm Michael P. michael.schramm@ag.tamu.edu
Texas A&M AgriLife Research, Texas Water Resources Institute, Texas A&M University , College Station , TX , United States of America
Latimer James
Electronic publication date: 2023 Sep 26
Publication date: 2023
Volume: 11
Electronic Location ID: e16073
Received 2023 Jun 29; Accepted 2023 Aug 20
Copyright: ©2023 Schramm
Copyright year: 2023
Copyright holder: Schramm
License: This is an open access article distributed under the terms of the Creative Commons Attribution License, which permits unrestricted use, distribution, reproduction and adaptation in any medium and for any purpose provided that it is properly attributed. For attribution, the original author(s), title, publication source (PeerJ) and either DOI or URL of the article must be cited.
License URL: https://creativecommons.org/licenses/by/4.0/

Keywords: Estuary, Water quality, Eutrophication, Texas, Generalized additive model, Nutrient loading

Funding: Texas Coastal Management Program grant approved by the Texas Land Commissioner The National Oceanic and Atmospheric Administration (NOAA), Office for Coastal Management NA21NOS4190136 This project was funded by a Texas Coastal Management Program grant approved by the Texas Land Commissioner, providing financial assistance under the Coastal Zone Management Act of 1972, as amended, awarded by the National Oceanic and Atmospheric Administration (NOAA), Office for Coastal Management, pursuant to NOAA Award No. NA21NOS4190136. There was no additional external funding received for this study. The funders had no role in study design, data collection and analysis, decision to publish, or preparation of the manuscript.

==============================
Evaluating estuary water quality responses to reductions (or increases) in nutrient loading attributed to on the ground management actions can be challenging due to the strong influence of environmental drivers on nutrient loads and non-linear relationships. This study applied generalized additive models to calculate watershed nutrient loads and assess responses in estuary water quality to seasonally-adjusted freshwater inflow and flow-adjusted nutrient loads in Lavaca Bay, Texas. Lavaca Bay is a secondary embayment on the Texas coast displaying early potential for eutrophication and water quality degradation. Use of flow-adjusted nutrient loads allowed the study to evaluate the response in water quality to changes in nutrient loads driven by anthropogenic sources. Cross-validation indicated that, despite data constraints, semiparametric models performed well at nutrient load prediction. Based on these models, delivered annual nutrient loads varied substantially from year to year. In contrast, minimal changes in flow-normalized loads indicate that nutrient loadings were driven by natural variation in precipitation and runoff as opposed to changes in management of nonpoint sources. Models indicated no evidence of long-term changes in dissolved oxygen or chlorophyll-a within Lavaca Bay. However, site specific long-term increases in both organic and inorganic nitrogen are concerning for their potential to fuel eutrophication. Further analysis found freshwater inflow had strong influences on nutrient and chlorophyll-a concentrations but there was no evidence that changes in watershed nutrient loading explained additional variation in dissolved oxygen and limited evidence that watershed nutrient loadings explained chlorophyll-a concentrations. In addition to providing a baseline assessment of watershed nutrient loading and water quality responses in the Lavaca Bay watershed, this study provides methodological support for the use of semiparametric models in load regression models and estuary assessments.

Introduction

Similar to many coastal areas globally, the coastal watersheds along the Texas Gulf coast are facing pressures from growing populations, increases in point source and non-point source pollution and alterations to freshwater flows that degrade water quality in downstream estuaries (Bricker et al., 2008; Kennicutt, 2017; Bugica, Sterba-Boatwright & Wetz, 2020). Despite these escalating pressures, national scale assessments have classified coastal estuaries in Texas as moderate or low risk for eutrophic conditions (Bricker et al., 2008). However, a suite of recent studies indicates that estuary water quality dynamics in both agricultural and urban dominated watersheds within Texas are expressing conditions that are increasingly conducive to algal blooms and eutrophication (Wetz et al., 2016; Wetz et al., 2017; Bugica, Sterba-Boatwright & Wetz, 2020; Chin, Beecraft & Wetz, 2022). With identification of several localized areas of estuary water quality concern along the Texas coast (Bugica, Sterba-Boatwright & Wetz, 2020), localized studies are being prioritized to better inform management actions.

The goal of this project is to assess watershed nutrient loading and the resulting water quality responses in Lavaca Bay, Texas. Lavaca Bay is a secondary bay in the larger Matagorda Bay system located roughly halfway between Houston, Texas and Corpus Christi, Texas. Lavaca Bay faces substantial challenges associated with legacy contamination but general water quality parameters such as dissolved oxygen (DO), nutrients, and biological parameters have been well within state water quality standards. Despite largely meeting state designated water quality thresholds, there have been concerning declines in abundance, biomass, and diversity of benthic fauna in Lavaca Bay (Beseres Pollack, Palmer & Montagna, 2011). These declines are partially attributed to reductions in freshwater inflow and changes in estuary salinity and are indicative of an already stressed system (Beseres Pollack, Palmer & Montagna, 2011; Palmer & Montagna, 2015; Montagna et al., 2020). More recently, significant linear increases in total phosphorus (TP), orthophosphate, total Kjeldahl nitrogen (TKN), and chlorophyll-a concentrations were identified at monitoring sites within Lavaca Bay (Bugica, Sterba-Boatwright & Wetz, 2020). While significant nitrogen increases were only identified in TKN, both organic and inorganic forms of nitrogen can serve as potential indicators of eutrophication status and estuary health (Jessen et al., 2015). Nitrogen is generally considered the limiting factor for primary production in many Texas estuaries (Gardner et al., 2006; Hou et al., 2012; Dorado et al., 2015; Paudel, Montagna & Adams, 2019; Wetz et al., 2017). However, estuaries can display nitrogen and phosphorus co-limitation that may vary temporally and spatially (Elser et al., 2007; Conley et al., 2009), emphasizing a need to assess a variety of nutrient species.

There are ongoing efforts between local, state, and federal agencies to address water quality impairments in the freshwater portions of the Lavaca Bay watershed (Jain & Schramm, 2021; Schramm et al., 2018; Berthold, Olsovsky & Schramm, 2021). However, on a statewide scale, these approaches have shown limited success and emphasize a need for improved methods to assess and link management actions with downstream water quality (Schramm, Gitter & Gregory, 2022). Such methods could help identify and replicate effective water quality management actions across the state . The identification and communication of changes and trends in water quality is complicated by the fact that trends are often non-linear and confounded by precipitation and runoff that hinder traditional analysis (Wazniak et al., 2007; Lloyd et al., 2014). The development and application of flexible statistical methods such as Weighted Regressions on Time, Discharge and Season (WRTDS, Hirsch, Moyer & Archfield, 2010) and Generalized Additive Models (GAMs, Wood, 2011) have provided effective tools for researchers to quantify and communicate non-linear changes in river and estuary pollutant loadings.

WRTDS calculates a time series of in-stream concentrations or loads (daily, monthly, or annually) and flow-normalized estimates of concentrations and loads using locally weighted regression for unique combinations of time, discharge, and season. WRTDS has been widely used to assess and identify trends in riverine nutrients (Oelsner & Stets, 2019; Stackpoole et al., 2021), chlorides (Stets et al., 2018), and other pollutants of concern (Shoda et al., 2019). WRTDS has also been successfully adapted to assess trends in estuary water quality concentrations (Beck et al., 2018).

While WRTDS is a statistical approach developed specifically for water quality applications, GAMs are a broadly applicable statistical method. GAMs are a semiparametric extension of generalized linear models where the response variable is modeled as the sum of multiple unknown smooth functions and parametric linear predictors (Wood, 2011). Although the underlying parameter estimation procedure of GAMs are substantially different than WRTDS, both the functional form and results have been demonstrated to be similar when assessing nutrient concentration trends (Beck & Murphy, 2017). Water quality applications of GAMs have included river and catchment nutrient concentration and load estimation (Wang, Kuhnert & Henderson, 2011; Kroon et al., 2012; Kuhnert et al., 2012; Robson & Dourdet, 2015; Hagemann, Kim & Park, 2016; McDowell et al., 2021; Biagi et al., 2022), and assessment of temporal trends of nutrients (Beck & Murphy, 2017; Murphy et al., 2019), phytoplankton (Bergbusch et al., 2021), and cyanobacteria (Hayes et al., 2020). Recently GAMs have also been used to link water quality responses in receiving water bodies to changes in nonpoint source nutrient inputs (Murphy et al., 2022). For a substantial discussion on the differences (and similarities) between GAMs and WRTDS for water quality applications readers are referred to Beck & Murphy (2017).

To provide actionable information for resource managers in Lavaca Bay, water quality conditions must be evaluated relative to changes in natural environmental drivers to better understand and manage potential human impacts. This study utilizes GAMs to develop estimates of delivered and flow-normalized nutrient loads and assess estuary water quality responses to changes in loads delivered to Lavaca Bay. GAMs were chosen over other regression-based approached for use in this study due to; (1) the ability to easily explore and incorporate different model terms; (2) the incorporation of non-linear smooth functions that do not require explicit a priori knowledge of the expected shape; and (3) inclusion of a link function that related the expected value of the response to linear predictors thus avoiding unneeded data transformations and bias corrections. The exploratory study also assesses the response of water quality parameters in Lavaca Bay over time and in response to freshwater inflow controlled for seasonality and to watershed nutrient loads that are controlled for environmentally driven variation.

Methods and Materials

Location and data

Lavaca Bay is a 190 km2 estuary with the majority of freshwater inflow provided by the Lavaca and Navidad River systems (Fig. 1). Lavaca Bay is relatively shallow with an average depth of 1.2 m and a generally well mixed and turbid water column (Beseres Pollack, Palmer & Montagna, 2011; Montagna et al., 2020). The Garcitas-Arenosa, Placedo Creek, and Cox Bay subwatersheds provide additional freshwater inflows. The proximity to freshwater inputs in the upper Lavaca Bay results in a strong mean salinity and variance gradient (Montagna et al., 2020). Average salinity ranges from 12 psu near the Lavaca River mouth to 23 psu at the mouth of Lavaca Bay. The variance is inversely correlated with mean salinity. The entire watershed area is 8,149 km2 and primarily rural. Watershed land cover and land use is 50% grazed pasture and rangeland, 20% cultivated cropland (primarily rows crops such as corn, cotton, and sorghum), and 5% suburban/urban. Pasture and rangeland is concentrated in the Lavaca River watershed, while cultivated crops are generally located along the eastern tributaries of the Navidad river. The Lavaca and Navidad River watersheds are a combined 5,966 km2, or approximately 73% of the entire Lavaca Bay watershed area. Discharge from the Navidad River is regulated by Lake Texana which has been in operation since 1980. Lake Texana provides 0.210 km3 of water storage and discharges into the tidal section of the Navidad River which ultimately joins the tidal section of the Lavaca River 15 km upstream of the confluence with the Lavaca Bay.

Figure 1 Map of Lavaca Bay watershed.

The freshwater sites are the most downstream freshwater stream locations with water quality and streamflow data used for nutrient load models. Water quality concentration data at the three Lavaca Bay sites were used to assess relationships between freshwater flows, loads and estuary water quality.

Daily discharges for the Lavaca River (USGS-08164000, Fig. 1) were obtained from the United States Geologic Survey (USGS) National Water Information System (https://waterdata.usgs.gov/nwis) using the dataRetrieval R package (De Cicco et al., 2022). Gaged daily discharges from the outlet of Lake Texana on the Navidad River (USGS-0816425) were provided by the Texas Water Development Board (TWDB) (April 21, 2022 email from R. Neupane, TWDB).

Water quality data for the two freshwater and three estuary locations were obtained from the Texas Commission on Environmental Quality (TCEQ) Surface Water Quality Monitoring Information System. Data submitted through the system are required to be collected under Quality Assurance Project Plans and lab method procedures outlined by the TCEQ’s procedures manual. These operating procedures ensure consistent collection and laboratory methods are applied between samples collected by different entities and under different projects. All sites had varying time periods and availability of water quality data. For freshwater locations, TP from January 2000 through December 2020 and nitrate-nitrogen (NO3) data from January 2005 through December 2020 were downloaded (Table 1). Less than 5-years of total nitrogen and TKN concentration data were available at the freshwater sites and deemed insufficient to develop load estimation models (Horowitz, 2003; Snelder, McDowell & Fraser, 2017). The three estuary sites included an upper-bay site near the outlet of the Lavaca River system (TCEQ-13563), a mid-bay site (TCEQ-13383), and the lower-bay site near the mouth of the Bay (TCEQ-13384). For estuary locations, we obtained data for TP, Nitrite+Nitrate(NOX), TKN, chlorophyll-a, and DO concentrations from January 2005 through December 2020 (Table 2). Freshwater nutrient samples were collected 0.3 m below the surface or halfway between the surface and bottom if depth was insufficient. Estuary nutrient samples were all collected 0.3 m below the surface. Estuary DO measurements were made at 3 locations along the water column (0.3 m, mid-depth, and bottom) and averaged. All water quality data was collected approximately quarterly.

Table 1 Summary of gauged streamflow and freshwater water quality samples between January 1, 2000 and December 31, 2020.

Station ID and watershed	Parameter	Mean	SD	N	Method	AWRLa	Criteriab	
USGS-08164000, Lavaca river	TP (mg/L)	0.21	0.09	80	EPA 365.1	0.06	0.69	
NO3 (mg/L)	0.18	0.24	74	EPA 353.2	0.05	1.95	
Mean daily streamflow (cfs)	332.78	1667.47	7671				
USGS-08164525, Navidad river	TP (mg/L)	0.20	0.08	81	EPA 365.1	0.06	0.20	
NO3 (mg/L)	0.29	0.26	62	EPA 353.2	0.05	0.37	
Mean daily streamflow (cfs)	666.14	2957.79	7671				
Notes.

a Ambient water reporting limit (AWRL) represents the highest concentration that can be used as a reporting limit by a laboratory for inclusion in the state’s monitoring data.

b The state of Texas has not adopted numeric criteria for nutrients, these criteria represent screening levels used by the state for assessment purposes.

Table 2 Summary of estuary water quality samples collected between January 1, 2005 and December 31, 2020.

Station ID	Parameter	Mean	SD	N	Method	AWRLa	Criteriab	
TCEQ-13563, Upper-Bay	TP (mg/L)	0.13	0.06	50	EPA 365.1	0.06	0.21	
NOx (mg/L)	0.09	0.13	53	EPA 353.2	0.05		
TKN (mg/L)	0.94	0.37	49	EPA 351.2	0.20		
Chlorophyll-a (ug/L)	9.67	5.33	49	EPA 445.0	3.00	11.60	
DO (mg/L)	7.91	1.34	56	ASTM D888-09(C) and TCEQ SOP V1		5.00	
TCEQ-13383, Mid-Bay	TP (mg/L)	0.11	0.05	47	EPA 365.1	0.06	0.21	
NOx (mg/L)	0.07	0.15	51	EPA 353.2	0.05		
TKN (mg/L)	0.94	0.49	45	EPA 351.2	0.20		
Chlorophyll-a (ug/L)	9.43	5.31	47	EPA 445.0	3.00	11.60	
DO (mg/L)	7.22	1.35	55	ASTM D888-09(C) and TCEQ SOP V1		5.00	
TCEQ-13384, Lower-Bay	TP (mg/L)	0.08	0.03	51	EPA 365.1	0.06	0.21	
NOx (mg/L)	0.06	0.08	52	EPA 353.2	0.05		
TKN (mg/L)	0.76	0.40	48	EPA 351.2	0.20		
Chlorophyll-a (ug/L)	8.22	6.44	46	EPA 445.0	3.00	11.60	
DO (mg/L)	7.51	1.32	54	ASTM D888-09(C) and TCEQ SOP V1		5.00	
Notes.

a Ambient water reporting limit (AWRL) represents the highest concentration that can be used as a reporting limit by a laboratory for inclusion in the state’s monitoring data.

b The state of Texas has not adopted numeric criteria for nutrients, these criteria represent screening levels used by the state for assessment purposes. NOx and TKN parameters are not currently screened by the state and do not have an associated screening level.

Estimating watershed based nutrient loads

Estimates of NO3 and TP loads at the Lavaca River (USGS-08164000) and the outlet of Lake Texana on the Navidad River (USGS-08164525) were developed using GAMs relating nutrient concentration to river discharge, season, and time. Separate models were fit at each station for each parameter and used to predict nutrient concentrations for each day in the study period. GAMs were fit using the mgcv package in R which makes available multiple types of smooth functions with automatic smoothness selection (Wood, 2011). The general form of the model related NO3 or TP concentration to a long term trend, season, streamflow, and two different antecedent discharge terms, shown in Eq. (1):

gμ=α+f1ddate+f2yday+f3log1+Q+f4ma+f5fa,

(1) y∼Nμ,σ2

where μ is the conditional expected NO3 or TP concentration, g() is the log-link, α is the intercept, fn() are smoothing functions. y is the response variable (NO3 or TP concentration) modeled as normally distributed with mean μ and standard deviation σ. ddate is the date converted to decimal notation, yday is numeric day of year (1–366), and Q is the mean daily streamflow.

Moving average (ma) is an exponentially smoothed moving average that attempts to incorporate the influence of prior streamflow events on concentration at the current time period (Wang, Kuhnert & Henderson, 2011; Kuhnert et al., 2012; Wang & Tian, 2013; Zhang & Ball, 2017), using Eq. (2): (2) mad=Qj+dQj−1+…+dj−1Q11+d+...+dj−1

where ma at discount factor d is calculated using flows (Q1, Q2, …, Qj) for days 1 through j. Here, as d approaches zero, the time series becomes the daily observed values and as d gets closer to one, the time series becomes the mean flow. Although Kuhnert et al. (2012) and Wang & Tian (2013) suggest using multiple covariates with d set at different values from 0.75 to 0.99, Robson & Dourdet (2015) and Zhang & Ball (2017) found substantial improvements in statistical nutrient models with d equal to 0.95 which was adopted in this study.

Flow anomaly (fa) is a dimensionless term that represents how wet or dry (or how anomalous) the current time period is from previous time periods (Vecchia et al., 2009; Zhang & Ball, 2017). Long-term flow anomaly (ltfa) is the streamflow over the previous year relative to the entire period (Eq. 3, Zhang & Ball, 2017) and and the short-term flow anomaly (stfa) calculated as the current day flow compared to the preceding 1-month streamflow (Eq. 4, Zhang & Ball, 2017):

(3) ltfat=x ¯1yeart−x ¯entireperiod,

(4) stfat=xcurrentdayt−x ¯1montht

where x are the averages of log-transformed streamflow over the antecedent period (1-year, 1-month, etc.) for time t. We used ltfa in NO3 models and stfa in TP models based on previous work demonstrating major improvements in NOX regression models that incorporated ltfa and moderate improvements in TP regression models that incorporated stfa (Zhang & Ball, 2017). Moving averages and flow anomalies were calculated with the adc R package (Schramm, 2023).

The calculation of model terms for the Lake Texana site were modified because daily loads are not a function of natural stream flow processes alone, but of dam releases and nutrient concentrations at the discharge point of the lake. Q, ma, and fa terms were calculated based on total gaged inflow from the 4 major tributaries to the lake. Thin-plate regression splines were used for ddate, log(1+Q), fa, and ma. A cyclic cubic regression spline was used for yday to ensure the ends of the spline match (day 1 and day 366 are expected to match). First order penalties were applied to the smooths of flow-based variables which penalize departures from a flat function to help constrain extrapolations for high flow measurements.

Left-censored data were not uncommon in this dataset (4% of TP samples and 20% of NO3 samples). Several methods are available to account for censored data. We transformed left-censored nutrient concentrations to one-half the detection limit. Although this simple approach can introduce bias (Hornung & Reed, 1990), we considered it acceptable because high concentrations and loadings are associated with high-flow events and low-flow/low-concentration events will account for a small proportion of total loadings (McDowell et al., 2021).

Daily loads were estimated as the predicted concentration multiplied by the daily streamflow. For the Navidad River (USGS-08164525) site, daily loads at the dam were calculated from the discrete daily concentration at the discharge point of the lake and corresponding reported daily discharge from the dam. Flow-normalized loads were estimated similar to WRTDS by setting flow-based covariates on each day of the year equal to each of the historical values for that day of the year over the study period (Hirsch, Moyer & Archfield, 2010). The flow-normalized estimate was calculated as the mean of all the predictions for each day considering all possible flow values. Standard deviations and 90% credible intervals were obtained by drawing samples from the multivariate normal posterior distribution of the fitted GAM (Wood, 2006; Marra & Wood, 2012; McDowell et al., 2021). GAM performance was evaluated with repeated 5-fold cross validation (Burman, 1989) using average Nash-Sutcliffe Efficiency (NSE), Pearson sample correlation (r) and percent bias (PBIAS) metrics.

Linking estuary water quality to hydrology and nutrient loads

To test if changes in freshwater inflow and nutrient loading had explanatory effect on changes in estuary water quality a series of GAM models were fit at each site relating parameter concentration to temporal trends (Eq. 5), temporal trends and inflow (Eq. 6), and temporal trends, inflow, and nutrient loads (Eq. 7): (5) gμ=α+f1ddate+f2yday,

(6) gμ=α+f1ddate+f2yday+f3Q,

(7) gμ=α+f1ddate+f2yday+f3Q+f4Load

where μ is the conditional expected response (parameter concentration), g() is the log link, and response variable was modeled as Gamma distributed with mean μ and scale λ. f1(ddate) is decimal date smoothed with a thin-plate regression spline, f2(yday) is the numeric day of year smoothed with a cyclic cubic regression spline. Prior work has shown that many water quality parameters may have lagged effects lasting days or even months following storms and large discharge events (Mooney & McClelland, 2012; Wetz & Yoskowitz, 2013; Bukaveckas et al., 2020; Walker et al., 2021). Walker et al. (2021) showed the impact of large flow events like Hurricane Harvey had somewhat short water quality impacts in Lavaca Bay and adjacent estuary systems, ranging somewhere around 20 days to a few months. We incorporate this lag effect by using cumulative totals for f3(Q) and f4(Load). f3(Q) is the cumulative of the previous 20 days of inflow (combined measurements from Lavaca River and Navidad River) and f4(Load) is the cumulative of the previous 20 days total NO3 or TP watershed load, both smoothed with thin plate regression splines. The set of models specified for each water quality response are in Table 3.

Table 3 Set of GAM models specified for each water quality parameter response.

Parameter	Model	Model terms	
TP	Temporal	s(ddate) + s(yday)	
Flow	s(ddate) + s(yday) + s(Q)	
Flow+Load	s(ddate) + s(yday) + s(Q) + s(TP Load)	
NOx	Temporal	s(ddate) + s(yday)	
Flow	s(ddate) + s(yday) + s(Q)	
Flow+Load	s(ddate) + s(yday) + s(Q) + s(NO3 Load)	
Chlorophyll-a	Temporal	s(ddate) + s(yday)	
Flow	s(ddate) + s(yday) + s(Q)	
Flow+Load	s(ddate) + s(yday) + s(Q) + s(TP Load) + s(NO3 Load)	
Dissolved Oxygen	Temporal	s(ddate) + s(yday)	
Flow	s(ddate) + s(yday) + s(Q)	
Flow+Load	s(ddate) + s(yday) + s(Q) + s(TP Load) + s(NO3 Load)	
TKN	Temporal	s(ddate) + s(yday)	
Flow	s(ddate) + s(yday) + s(Q)	

Because streamflow and nutrient loads are tightly correlated, freshwater inflow can mask signals due to changes in nutrient loads alone. Following the methodology implemented by Murphy et al. (2022), both freshwater inflow and nutrient loads were prepossessed to account for season and streamflow respectively. Cumulative inflow values were replaced by seasonally adjusted cumulative inflow obtained from the residuals of a GAM model fit between season (day of year) and log transformed cumulative inflow. Nutrient loads utilized the flow-normalized loads estimated in the previous section.

This study used an information theoretic approach to evaluate evidence of model covariate effects on Lavaca Bay water quality. Model probabilities were calculated and compared using the AICc scores between each group of temporal, inflow, and inflow+load models (Burnham, Anderson & Huyvaert, 2011). Improvements in model probabilities provide evidence that the terms explain additional variation in the response variable.

Results

Watershed nutrient loads

Predictive performance of nutrient loads ranged from “satisfactory” to “very good” based on standardized evaluation metrics of NSE, r, and PBIAS (Moriasi et al., 2015) calculated using 5-fold cross validation. Median goodness-of-fit metrics for NO3 models in the Lavaca River were 0.34 NSE, 0.70 r, and 2.00 PBIAS. Navidad River NO3 models appeared to perform slightly better with 0.48 NSE and 0.87 r but with higher bias at 10.90 PBIAS. Generally, TP models performed better than NO3 models. Median goodness-of-fit metrics for TP in the Lavaca River were 0.81 NSE, 0.93 r, and −7.20 PBIAS. Navidad River TP models had similar performance with 0.91 NSE, 0.99 r, and −3.30 PBIAS. Density plots of metrics show similar distribution of values between sites for the same parameter, with the exception r values for NO3 loads where Lavaca River had a much larger variance in values compared to the Navidad River (Fig. 2). TP GAMS had higher average NSE and r values and lower variance in metric values compared to NO3.

Figure 2 Density plots of goodness-of-fit metrics (NSE, r, and PBIAS) from repeated 5-fold cross validation between predicted nutrient loads from GAM models and measured nutrient loads.

Color indicates the tail probability calculated from the empirical cumulative distribution of the goodness-of-fit metrics. Values closer to 1 for NSE and r and values closer to 0 for PBIAS represent more ideal goodness-of-fit assessments.

Annual NO3 and TP loads show considerable variation, generally following patterns in discharge (Figs. 3, 4). Flow-normalized TP loads at both sites and flow-normalized NO3 loads in the Lavaca River indicated watershed based loads did not change much over time when accounting for variation driven by streamflow (Fig. 3). Flow-normalized loads in the Lavaca River showed small variation over time with some decreases in NO3 loads since 2013.

Figure 3 Aggregated estimated annual and flow-normalized annual NO3 and TP loads for the Lavaca (USGS-08164000) and and Navidad (USGS-08164525) Rivers.

Figure 4 Comparison of delivered annual loads and annual discharge at the Lavaca (USGS-08164000) and Navidad (USGS-08164525) Rivers.

Aggregated across both sites, the mean annual NO3 load from 2005 through 2020 was 205,405 kg (126,867 kg–341,569 kg, 90% CI). Annual NO3 loads ranged from 12,574 kg in 2011 to 794,510 kg in 2007. Total annual TP loads ranged from 7,839 kg in 2011 to 595,075 kg in 2007. Mean annual TP loading from 2005 through 2020 was 182,673 kg (152,227 kg–219,310 kg, 90% CI). On average, the Navidad River accounted for 68% of NO3 loads and 59% of TP loads from 2005 through 2020. However, during periods of extreme drought the Lavaca River became the primary source of nutrient loading in the watershed with the Navidad River only accounting for 15% and 25% of NO3 and TP loads in 2011 (Fig. 4).

Linkages between water quality and watershed flows and loads

There was no evidence of long-term changes in TP or DO concentrations at any Lavaca Bay site (Fig. 5). The upper-bay site(TCEQ-13563) had evidence of a long-term linear increase in NOX while chlorophyll-a decreased from 2005 through 2014 (Fig. 5). NOX concentration at the mid-bay site (TCEQ-13383) displayed a periodic pattern that is indicative of a strong influence from inflow or precipitation. The temporal GAMs did not provide evidence of long-term trends in any of the water quality constituents at the lower-bay site (TCEQ-13384).

Figure 5 Fitted splines (shaded regions indicate 90% confidence intervals) from the temporal estuary GAM (Table 3) display the marginal smoothed effect of date on TP (A), NOX (B), chlorophyll-a (C), TKN (D), and DO (E) concentrations at each site in Lavaca Bay.

Freshwater inflow provided additional explanation for changes in NOx and chlorophyll-a at all of the Lavaca Bay sites according to AICc and model probability values (Table 4). Freshwater inflow also explained additional variation in TP at the upper- and mid-bay sites but not the lower-bay site. Freshwater inflow did not explain additional variation in TKN concentrations at any of the sites. The upper- and lower-bay sites saw improvements in explanation of DO concentration with the inclusion of inflow.

Inclusion of TP loads provided additional explanation of TP concentrations at the upper- and mid-bay sites. Inclusion of NO3 loads did not provide model explanatory improvements at any of the sites. The addition of nutrient loads (both TP and NO3) terms did not provide additional explanation for changes in DO concentrations but did provide model improvement for the chlorophyll-a model at the mid-bay site.

Increases in aggregated freshwater inflow resulted in increases in TP at the upper- and mid-bay sites and increases in NOX and chlorophyll-a concentration at all three sites (Fig. 6). Linear increases in DO concentration were observed with increasing flow at the upper- and lower-bay sites. TKN showed no response to changes in inflow at any of the sites.

Increased TP loads resulted in nearly linear increases of TP concentration at the upper- and mid-bay sites (Fig. 7). The relative effect size appeared much smaller than the effect of freshwater inflow alone. Increased NO3 loads did effect NOx concentrations at any site. Linear increases in chlorophyll-a were observed in response to increased TP loads at the mid-bay site, but not the other sites or in response to NO3 loads.

Table 4 Estuary GAM AICc values and associated model probabilities (in parenthesis). Models with the highest probability for each site and water quality parameter combination are bolded and italicized for emphasis.

Parameter	Site	Temporal	Inflow	Inflow + Load	
TP	Upper-Bay	−145.3 (0.01)	−150 (0.12)	−154 (0.87)	
Mid-Bay	−152.1 (0.02)	−155.1 (0.07)	−160.2 (0.91)	
Lower-Bay	−194.4 (0.39)	−194 (0.31)	−194 (0.31)	
NOx	Upper-Bay	−175.1 (0.01)	−183.8 (0.5)	−183.8 (0.5)	
Mid-Bay	−218.9 (0)	−241.7 (0.5)	−241.7 (0.5)	
Lower-Bay	−263.4 (0)	−298.1 (0.5)	−298.1 (0.5)	
Chlorophyll-a	Upper-Bay	289.5 (0.37)	289.5 (0.38)	290.3 (0.25)	
Mid-Bay	279.7 (0.24)	279.6 (0.26)	278.2 (0.5)	
Lower-Bay	268.2 (0.03)	262.7 (0.48)	262.7 (0.48)	
TKN	Upper-Bay	31.1 (0.5)	31.1 (0.5)	–	
Mid-Bay	42.2 (0.5)	42.2 (0.5)	–	
Lower-Bay	34.3 (0.5)	34.3 (0.5)	–	
DO	Upper-Bay	138.3 (0.17)	136.4 (0.42)	136.4 (0.42)	
Mid-Bay	146.4 (0.37)	146.8 (0.29)	146.5 (0.34)	
Lower-Bay	135.9 (0.04)	130.6 (0.48)	130.6 (0.48)	

Figure 6 Fitted splines from estuary GAMs display the marginal smoothed effect of freshwater inflow (controlled for season) on TP (A), NOX (B), chlorophyll-a (C), TKN (D), and DO (E) concentrations at each site in Lavaca Bay.

Figure 7 Fitted splines from the nutrient loading GAMs display the marginal smoothed effect of 20-day aggregated TP and NO3 flow-normalized loads on (A) TP, (B) NOX, and (C, D) chlorophyll-a concentrations at each site in Lavaca Bay.

Discussion

Nutrient loads

TP and NO3 loadings from the Lavaca Bay watershed showed high inter-annual variability driven primarily by fluctuations in discharge. Notably, there were no indications of trends in flow-normalized NO3 and TP loads in the Navidad River. In comparison, there was weak evidence of more recent decreases in flow-normalized NO3 (but not TP) loads in the Lavaca River watershed. While the dominant agricultural land uses differ between the Lavaca (primarily grazed pasture and rangeland) and Navidad(mix of pasture and row crops) catchments, we did not have a reason to expect different flow normalized trends between the two systems from land use alone. Freshwater discharges in the Navidad River are regulated by the Palmetto Bend Dam forming Lake Texana at the lower extent of the river. Lentic nitrogen uptake and cycling may have regulating effects that mask changes in upstream nitrogen loadings. Additional nutrient data collection in the tributaries of Lake Texana is needed to fully assess the role of Lake Texana in regulating nutrient delivery to the Lavaca Bay system. However, these results suggest that there have been no changes in the NO3 or TP loading from the Navidad River system at the Lake Texana discharge point when accounting for variations in year to year discharge.

The evidence of decreased Lavaca River NO3 loading, although weak, is a potential positive sign for water quality managers working to implement practices that improve water quality in the freshwater sections of the Lavaca River watershed. Planning and implementation efforts to increase agricultural producer participation in water quality protection practices have been ongoing in the watershed since 2016 (Schramm et al., 2018; Berthold, Olsovsky & Schramm, 2021), however little work has been conducted to directly link these efforts with water quality outcomes. The decrease in flow-normalized NO3 loads could be a reflection of those collective efforts but the lack of evidence for similar changes in flow-normalized TP loads provide contrary support. The inconsistent flow-normalized trends may also reflect some of the weakness of the water quality dataset that is primarily composed of ambient water quality measurements. The issues associated with the lack of flow-biased measurements is further discussed later in this section.

Some prior studies have generated estimates of mean annual TP yields in the Lavaca River watershed (Table 5, Dunn, 1996; Rebich et al., 2011; Omani, Srinivasan & Lee, 2014; Wise, Anning & Miller, 2019). Although these studies differ in time periods and methodologies, they provide a sanity check for the reasonableness of the annual estimates generated in the current study. In a regional assessment of nutrient loading in river’s along the Gulf of Mexico, Dunn (1996) used the LOADEST model to develop an estimated mean annual yield of 28.9 kg/km2. LOADEST is a multiple linear regression model that fits log transformed pollutant concentrations to long term, seasonal, and flow based predictors and includes methods for bias correction when exponentiating the response variable. Rebich et al. (2011) and Wise, Anning & Miller (2019) used SPARROW to provide a more recent assessment of regional catchment based loadings to the Gulf of Mexico(Table 5). SPARROW is a hybrid statistical-process model with the underlying nutrient load estimation methods based on the previously described LOADEST (Schwarz et al., 2006). The functional form of the LOADEST regression model is similar to the terms applied in the GAMs used in the current study. The only study to apply a mechanistic watershed model (SWAT) to estimate nutrient loadings in the Lavaca River watershed Omani, Srinivasan & Lee (2014) developed estimated yields (42 kg/km2) similar to the two SPARROW models. Although direct comparisons are complicated by varying time periods, the estimates in this study do fall within the range of of previously developed estimates. To evaluate changes in long-term trends in discharge might be associated with the nutrient yield estimates covering different time periods, we fit a GAM relating log-transformed daily discharges on the Lavaca River to season and time (Fig. 8). The long-term trends in discharge indicate watershed discharges were at or above average from 1972 though the early and mid-1980s. In comparison watershed discharges since the mid-2000s are at or below average. It is probable that the lower than average discharges observed from 2010 through 2021 (Fig. 8) bias our estimates downward compared to studies that included higher than average streamflow periods (1995–2005). Overall, the ranges of estimated yields among different studies along with the apparent large variability in streamflow driven loadings (Fig. 3, Fig. 4) suggest that the current estimates of TP loading are reasonable.

Table 5 Comparisons of previously published estimates of mean annual TP yield at the Lavaca River site.

Reported yield (kg km2 year−1)	Approach	Time period	Reference	
35.2 (28.8, 43.3)a	GAM	2005–2020	This work	
45.2	SPARROW	2000–2014		
42	SWAT	1977–2005		
20.81–91.58b	SPARROW	1980–2002		
28.9	LOADEST	1972–1993		
Notes.

a Mean of the annual point estimates and the lower and upper 90% credible intervals.

b Represents a binned value range from a choropleth map.

Figure 8 Measured daily discharges (log-transformed) and smoothed long-term trends for the Lavaca River form 1972 though 2001.

Estuary water quality

The non-linear estuary water quality trends identified in the current study differed slightly from previously identified trends (Bugica, Sterba-Boatwright & Wetz, 2020). This is not unexpected due to the different time periods, different methodology, and generally small slopes identified for most of the significant water quality parameters in prior work. Both DO and cholorophyll-a concentrations at all three Lavaca Bay sites were stable from 2005 through 2020. This is a positive outcome in comparison to other Texas estuaries that are facing larger demands for freshwater diversions, higher population growth, and more intense agricultural production which have resulted in more direct signs of eutrophication (Wetz et al., 2016; Bugica, Sterba-Boatwright & Wetz, 2020). Despite the stability of DO and cholorophyll-a, there are concerning site specific increases in NOX and TKN concentration over the same time period. These trends are especially concerning due to the nitrogen limitation identified in many Texas estuaries (Gardner et al., 2006; Hou et al., 2012; Dorado et al., 2015; Paudel, Montagna & Adams, 2019; Wetz et al., 2017) and the relatively low ambient concentrations observed in Lavaca Bay.

The strong positive effect of freshwater inflow on NOX and TP concentration are suggestive of nonpoint watershed sources, consistent with watershed uses and with other studies relating freshwater inflow with nutrient concentrations in Lavaca Bay and other estuaries (Russell, Montagna & Kalke, 2006; Caffrey et al., 2007; Peierls, Hall & Paerl, 2012; Palmer & Montagna, 2015; Cira, Palmer & Wetz, 2021). Inflow had a non-linear relationship with NOx, with NOx increasing as freshwater inflow transitioned from low to moderate levels. At higher freshwater inflows, the NOx decreased at the mid- and lower-bay sites, possibly indicating a flushing effect at higher freshwater inflow. No relationship between TKN and freshwater inflow was observed at any of the sites. The results coincide with previous work that suggest processing of organic loads in the tidal portions of the Lavaca River reduce transport of nutrients to the lower reaches of Lavaca Bay (Russell, Montagna & Kalke, 2006).

Freshwater inflow also displayed positive effects on chlorophyll-a at each site, with the largest effect at the lower-bay site. The lower-bay site also showed an attenuation in in chlorophyll-a at the highest freshwater inflow volumes. Freshwater flushing or increases in turbidity are associated with limitations or decreases in chlorophyll-a in other estuaries (Peierls, Hall & Paerl, 2012; Cloern, Foster & Kleckner, 2014). No relationships between inorganic nitrogen and chlorophyll-a were observed. However, a small positive effect between flow-normalized TP loads and chlorophyll-a concentrations were detected at the mid-bay site. Although, this is not suggestive of a phosphorus limitation in the Bay, it is supportive of prior work emphasizing the importance of controlling both nitrogen and phosphorus runoff to protect water quality (Conley et al., 2009). Due to the lack of TKN loading information, no assessment between organic nitrogen loads and chlorophyll-a were possible.

Although other studies have identified complex relationships between estuary nutrient concentrations, nutrient loading and chlorophyll-a concentrations in Texas estuaries (Örnólfsdóttir, Lumsden & Pinckney, 2004; Dorado et al., 2015; Cira, Palmer & Wetz, 2021; Tominack & Wetz, 2022), this study specifically used flow-adjusted freshwater derived nutrient loads to parse out contributions from changes in nutrient loadings while accounting for variations in load due to flow. Loading GAMs indicated no evidence of changes in flow-normalized TP loads in either river, and no changes in flow-normalized NO3 loads in the Navidad River. The small changes in flow-normalized NO3 loads in the Lavaca River are probably masked under most conditions by discharge from the Navidad River. Given the relatively small variation in flow-normalized loads, it can be expected that they would contribute little to the variance in downstream water quality.

There was no evidence that adjusted freshwater inflow and nutrient loads had effects on DO concentration in Lavaca Bay. The seasonality term in the temporal GAM models explained a substantial amount of DO variation at all of the sites. Responses of estuary metabolic processes and resulting DO concentrations can be quite complicated and often locally specific (Caffrey, 2004). While the lack of total nitrogen or TKN loading data hinders interpretation, the large seasonal effect on DO concentration indicates physical factors (such as temperature, wind, and turbidity) play an important role and should be included in future models. Prior work suggests that Lavaca Bay may not be limited by nutrients alone, with high turbidity or nutrient processing in upper portions of the Bay or intertidal river limiting production (Russell, Montagna & Kalke, 2006). Finally, it is reasonable to assume that fluctuations in DO may not occur immediately in response to nutrient pulses or freshwater inflow. Work has shown that many water quality parameters may have lagged effects lasting days or even months following storms and large discharge events (Mooney & McClelland, 2012; Wetz & Yoskowitz, 2013; Bukaveckas et al., 2020; Walker et al., 2021). However, this study only evaluated responses to 20-day cumulative loading and inflows.

Overall, this study suggests that DO and chlorophyll-a concentrations have been relatively stable in Lavaca Bay. Site-specific increases in TKN and NOX concentrations are a cause of concern for increasing risks of eutrophication within Lavaca Bay which might be currently attenuated by changes in freshwater flow, turbidity, and other physical processes. While loading models indicate that there are large annual fluctuations in NO3 loads, these changes have been largely driven environmental conditions (changes in runoff and river discharge). These models also provide evidence that estuary NOX and TP concentrations are strongly driven by freshwater inflow and to a lesser extent fluctuation in flow-normalized riverine loadings. Site-specific changes in the relationships between freshwater inflow and responses in both chlorophyll-a and NOx concentrations are indicative of nutrient processing and or tidal flushing effects moving from the river discharge point to the mouth of Lavaca Bay. This study does not completely explain site specific increases in NOX and TKN concentrations in Lavaca Bay. The freshwater study sites did not quantify nutrient loadings from tidal contribution areas or ungauaged watersheds. Nutrient contributions from wastewater facility discharges, septic systems, and stormwater could be considerable contributors to nutrient loadings in Lavaca Bay since they are not processed by a tidal river reach prior to entering the Bay. The Garcitas-Arenosa Creek, Placedo Creek, and Cox Bay subwatersheds are currently undersampled but compose approximately 27% of the watershed area. The contribution of nutrient loadings from these undersampled areas is unknown.

Future management and research needs

The GAM approach proved useful for both estimating loads and assessing downstream responses in water quality. Although we did not compare other models, it is likely similar estimates of loadings would be obtained by methods such as LOADEST, WRTDS, or SPARROW given the functionally similar dependent variable structures. The underlying weakness in the estimates of loading in the current study is the reliance on ambient water quality data used for statewide water quality assessments. Cross-validation of the nutrient loading models highlights that predictions are prone to high bias, owing to the lack of targeted storm or flow biased measurements. The high biases are indicative that subsets of values were unable to capture the functional relationships with the flow based dependent variables. It was beyond the scope of the current study to evaluate the subsets of cross-validation data and scores. However, the cross-validation procedure is indicative that more robust sampling is needed. Supplementary flow-biased monitoring targeting storm- or high-flow conditions is critical to improve model performance and strength of evidence produced by these models (Horowitz, 2003; Snelder, McDowell & Fraser, 2017). Although there is existing work on the samples sizes and sample design required for reliable performance of both LOADEST (Park & Engel, 2014) and WRTDS (Kumar et al., 2019) models, similar work does not appear to have been extended to water quality applications of GAMs.

Due to the concerning increases in eutrophication associated parameters in Lavaca Bay and other Texas estuaries (Bugica, Sterba-Boatwright & Wetz, 2020), and the desire to quantify linkages between environmental outcomes and on the ground management actions (Schramm, Gitter & Gregory, 2022) there is a strong need for reliable estimates of pollutant loadings and responses along the Texas coast. Within Texas, statewide water quality monitoring programs have focused on collection of ambient condition data. A framework for establishing pollutant load monitoring programs across catchments that explicitly incorporate flow biased data is needed for assessing nutrient loading and estuary health along the Texas coast.

Additional efforts focused on identifying relevant effect sizes, sampling designs, and funding mechanisms that can support long term efforts are also needed to adequately design such a framework. Large long-term monitoring programs in and around the Chesapeake Bay, San Francisco Bay, and along the Mississippi River have proven extremely effective at informing management actions and tracking progress towards long-term pollutant reduction goals. Similar coordinated efforts across Texas coastal watersheds would prove useful for resource management efforts intended to protect the biological and water quality integrity of Texas’s estuaries.

Conclusions

The primary purpose of this study was to provide estimates of watershed nutrient loadings and assess water quality responses to changes in nutrient loads. GAMs provided reliable estimates of watershed NO3 and TP loads. However, additional flow-biased data collection efforts are needed to decrease the prediction variance and improve accuracy at critical high-flow loading events. While some ongoing projects will fill data gaps for total nitrogen and TKN loading, additional efforts are needed to coordinate data collection efforts specifically for load estimation across Texas estuaries. Despite these data gaps, this study identified high annual fluctuations in nutrient loads driven primarily by discharge. No evidence was identified to indicate that on the ground management had changed nutrient loading in the Navidad River subwatershed. There was weak evidence for recent reductions in flow-normalized NO3 loading in the Lavaca River subwatershed although the results are at odds with flow-normalized trends in TP loads.

This study, consistent with others along the Texas coast, found strong effects of freshwater flow on nutrient and chlorophyll-a concentrations. DO concentrations, dominated by seasonal effects, did not show strong direct responses to freshwater flow. Small variances in flow-adjusted nutrient loads indicate that (1) there have been limited changes in non-point sources of nutrients and (2) that there is not strong evidence that those small changes have had extensive effects on chlorophyll-a or dissolved oxygen in Lavaca Bay. Although this study did not identify changes in DO or chlorophyll-a concentrations in Lavaca Bay, site specific increases in NOX and TKN are a cause for water quality concern. The study provides a baseline assessment for future water quality management activities in the watershed. In order to effectively track and link improvements or degradation of water quality conditions in Lavaca Bay and other coastal Texas watersheds with on the ground efforts, more robust sampling networks are needed to improve spatial coverage of undersampled areas and explicitly incorporate flow-biased sampling.

The author thanks Stephanie deVilleneuve for comments on an early draft and members of the Coastal Nutrient Input Repository project committee for supporting project development and their valuable input. The views expressed herein are those of the author(s) and do not necessarily reflect the views of NOAA, the US Department of Commerce, or any of their subagencies.

Additional Information and Declarations

Competing Interests

Author Contributions

Data Availability

The authors declare there are no competing interests.

Michael P. Schramm conceived and designed the experiments, performed the experiments, analyzed the data, prepared figures and/or tables, authored or reviewed drafts of the article, and approved the final draft.

The following information was supplied regarding data availability:

The reproducible code and datasets generated during this study are available at Zenodo:

Michael, Schramm. (2023). TxWRI/lavaca-nutrients: Texas Coastal Nutrient Input Repository - Lavaca Bay (v1.3). Zenodo. https://doi.org/10.5281/zenodo.7630758.

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
