# Peer review of "Linking watershed nutrient loading to estuary water quality with generalized additive models"

_PeerJ, doi:10.7717/peerj.16073_

## Round 0.1 · original submission · Major Revisions

Although 2 of the 3 reviewers indicated only minor revisions, the third had some suggestions that were more substantial. All 3 said the paper was very good so when you make the adjustments that the reviewers suggest you should be good to go.

Reviewer 1 ·

Basic reporting

Abstract
Very well-structured abstract – introduced the problem, the objectives of the research, the results obtained, and the conclusion. No changes are recommended.

Introduction
The introduction was well-structured and easy to follow. No changes are recommended.

Experimental design

Line 114: A link to the USGS NWIS system could be added
A brief description of the sample collection and laboratory analyses described in the QAPP and TCEQ procedures manual, detection and recovery limits, and instrumentation would seem appropriate.
Also, what are the state-recommended guideline values? Include them in Table 2

Validity of the findings

Results
No comments

Discussion
Line 371 “has” repeated twice

Additional comments

The author adequately answered the research questions and fulfilled all the stated objectives. The results from the GAM model provided estimates of watershed nutrient loadings and an outlook of nutrient loading's influence on water quality.
The whole of the paper was well structured, easy to read, and contributes sufficiently to the existing body of knowledge on estuary anthropogenic pollution and water quality monitoring.
Through this paper, the author has created an addition to the body of work on using models for the prediction and assessment of pollution in estuaries and aquatic bodies, as a means of recommending and establishing policy actions for pollution mitigation.

·

Excellent Review

This review has been rated excellent by staff (in the top 15% of reviews)
EDITOR COMMENT
This reviewer should be commended for this very thorough review. It required significant effort and time. The paper will be better due to it. Thank you.

Basic reporting

The writing, references, article structure, and hypotheses/results are sufficient. See my comments under "general comments" for more information.

Experimental design

See my comments under "general comments". I have some concerns about the modeling of the estuarine response variables.

Validity of the findings

The GitHub repo linked through Zenodo is excellent, I always appreciate when authors provide reproducible code.

Additional comments

This article describes the use of Generalized Additive Models (GAMs) to estimate nutrient loads entering Lavica Bay from the main inflows. The author uses these models to evaluate the anthropogenic contribution of nutrients to the bay through flow-normalization and to evaluate if these loads are related to long-term variation in estuarine water quality. Overall, this is a valuable paper that further validates the use of GAMs for long-term trend analysis and will have particular value for the management of this system. My comments below relate to either clarification or adjustment of the methods that authors should consider. If pursued, I consider these changes appropriate for a major revision.

I have some concerns about the methods starting on line 184 that describe how the load estimates are related to the estuarine data. First, I strongly encourage the authors to develop these models using cumulative or lagged estimates for the flow and load variables. It appears that these are currently modeled as the estimates for the same day the water quality parameter was sampled. I would not expect much of a relationship between water quality and the independent variables at that time scale. I suspect that the relationships the author shows regarding flow/load with the response reflects some level of temporal autocorrelation between the daily estimates and preceding conditions/loads, where the latter may be a truly causative relationship. I see two potential approaches to remedy the issue. One, the author could test different lagged values for flow and load, although this would require a bit of trial and error. A potentially better solution is to use the aggregate flow volume and load between the sample dates as a sum of the volume and mass, respectively, between each observation of the time series for the response variable. I would expect these aggregate measurements to be much more strongly correlated to the response variables than the daily values that are currently used. This limitation is noted on lines 371-374, so I’m curious why the authors did not use this approach.

My second concern about the models in eqns. 5-7 also relates to the flow and load variables. It’s stated on line 199-201 that flow-adjusted load values were used, which makes sense, but I don’t understand how this additional analysis differs from the flow-normalization in the previous section (lines 176 – 179). Why not just use the flow-adjusted loads from the previous models? Once this is clarified (or corrected), it might make more sense to change eqn 6 to include smoothers for ddate, yday, and flow-normalized load, and eqn 7 to included smoothers for ddate, yday, and load. These wouldn’t be “nested” models for AIC comparison, but would perhaps provide a better approach to evaluate loads related to natural or non-natural variation. I don’t see the value in including flow as a separate variable since it’s basically a surrogate for total load.

I also strongly encourage the author to consider replacing “semiparametric models” with “Generalized Additive Models” in the title. It’s a more accurate description of what’s included in the paper.

Minor comments:

Line 18: Maybe replace “non-natural variation” with “anthropogenic sources”.

Line 61: Accidental line break.

Line 77: Suggest rephrasing as “…generalized linear models where the response variable is modeled as the sum of…”

Tables 1 and 2: It would be useful to indicate the approximate sampling interval for the water quality parameters. It looks like quarterly.

Line 102: Some more information about the Bay would be helpful. What is the average depth? Any information on residence time? Is the water column mixed, stratified, or seasonally-stratified?

Lines 128 – 129: The interpretation of the figures, tables, and text would be much easier if the estuarine sites were simply referred to as “upper”, “mid”, and “lower”. I suggest replacing all station names with these terms throughout the manuscript.

Line 129 – 131: Some additional information about the sampling methods for the estuarine sites could be useful. Where in the water column were these samples taken? Is the water column mixed? I’m thinking specifically about dissolved oxygen. The results show that it was not related to load, but you may have different outcomes depending on surface vs bottom DO.

Line 138: Typo “tend”.

Figure 1: Remove “contribution” from the legend.

Eqn 2: I think some additional details are needed in the text below this equation. Why was d set at 0.95? Based on the text, this seems awfully close to 1 or the “mean flow”. Also, the equation includes a lag term, t – t_j, but the size of this lag is not stated, i.e., “some historical day of observation”. Was this set explicitly by the author, and if so, is there a justification?

Line 151: It’s not clear to me how fa is a unitless term. The equations show these as simple differences of average streamflow, so wouldn’t the units be in log-cfs?

Line 169: What percentage of the observations for each parameter were non-detects? I realize that setting the values as ½ the limit can introduce bias, but this may not be a big deal if there aren’t many non-detects. There does appear to be a GAM package that handles non-detects with Tobit models, but it is quite old (by software standards). It may be worth exploring. https://cran.r-project.org/web/packages/cenGAM/index.html

Line 236: This might be a personal preference, but “unusual periodic pattern” seems like regular periodicity and not a more random periodicity as one would expect with rain events. Consider rephrasing this text.

Figure 2 caption: Consider adding a sentence like “Values closer to 1 for NSE and r and values closer to 0 for PBIAS represent more ideal goodness-of-fit assessments.”

Figure 3 caption: Consider adding Lavaca and Navidad with the station name, as in the caption for figure 4.

Figure 5 caption: Consider including a reference to Table 3, as it wasn’t immediately clear to me that these models didn’t include flow or load.

Table 4: As for my previous comment, I’d replace the site names with “upper”, “mid”, and “lower” and arrange them accordingly. The current arrangement is “mid”, “lower”, and “upper”. Also, it looks like several of the statistical summaries are identical, e.g. the inflow and inflow+load model for TP at TCEQ-13384. Are these simply rounding issues? And if so, are these models really all that different in terms of explained variance?

Line 272: The text “contributing tributaries” seems redundant. Maybe remove “contributing”.

Line 341: Replace “were” with “was”.

Line 395: Replace “suggest” with “such as”.

Line 420: Replace “provide” with “prove” or “be”.

Reviewer 3 ·

Basic reporting

This manuscript was written clearly and I was pleased that I had no trouble understanding the author's message. The author's use of literature citations are thorough and support the reasoning for the study and the methods used. I thought all the figures and tables were useful for demonstrating the methods and findings. I only have a few clarifying comments on basic reporting:

Table 1: It would be helpful if you would include the watershed or station name under the USGS station ID in this table.

Line 164: Not sure if this is a typo, or if I’m missing what log1p(Q) is.

Line 392, "Limitation" section: I expected this to be a discussion of limitations of the approach like matching daily loads to concentrations, the complex environmental factors that could be involved in chlorophyll and DO trends, changing climate, etc. However, since you focus on the (very valid) problem of limited sampling, you may want to rename this sub-section of the Discussion. Maybe call it "Data Needs" or "Suggested Management Focus" or something like that?

Experimental design

The need for the research and methods were well defined. My last two comments here (Lines 194 and 362) are the ones where I think the methods could be expanded/modified. However, the study is useful as-if, so if the editors and author would rather these suggestions be designated as "future work", that seems reasonable as well.

Line 53-54: Since only one of the trends you mention from Bugica is for a nitrogen species, I suggest rephrasing or expanding this sentence to explain why we’d be concerned about all of the parameters that are increasing (or clarify if we are only concerned about the TKN).

Lines 129-131: It would be helpful if you described how deep the water column is at the estuary sample locations and from what depth the samples that you analyzed were taken. A little more explanation of the estuary dynamics would be helpful too -- is this a stratified or well-mixed system, what is the range in salinity, is there a key season of concern for eutrophication?

Lines 194-201: Some discussion is needed of how you dealt with temporal matching of flow and loads to concentrations in the estuary. I see in the discussion (line 373-374) that you matched flow and load estimated for the day of each sample. This approach should be explained in the methods, and I think you need some explanation is needed as to why that would be appropriate and/or how it may impact the results. As you discuss below, you might get better relationships (for Table 4, Lines 244-248) if you either lagged the match or matched an average load and flow from some reasonable period before the estuary sample date. If you had the ability to try some models like that, I think it may be a useful addition.

Lines 362-374: Is low oxygen a year-round problem in similar systems in Texas, or just summer? I ask because I wonder if you should pull out just the summer oxygen and see if there is more of signal to analyze with relationship to the watershed. Alternatively, an interactive term in the GAM between season and flow might be interesting to try if the response to flow only happens at certain times of year. At a minimum, I think the manuscript should mention that specific seasonal signals may be lost in the annual analysis and future work could include more detailed look at a particular season.

Validity of the findings

The findings are discussed thoroughly and explained well. My few comments related to conclusions and discussion are:

Line 296: Would damming of the Navidad River impact the loads through the Lavaca River watershed? I don’t see why if the data was from the USGS-08164000 site.

Lines 286-321: I think this discussion is too long comparing the TP yield first to the Dunn study, and then to 3 other studies. The TP yield estimate from your study is in the middle of the others, so you can condense this 2 paragraph discussion and simply say that you fall in the middle of other estimates and state briefly that there are reasons such as land use and flow variability that the estimates vary.

Lines 360-361: It is probably worthwhile to note that the effect of flow you see in chlorophyll is very likely due to more nutrients coming in with high flows. I wouldn’t want a reader to think that since your flow-adjusted nutrient loads are not linked to chlorophyll that this means there is definitely no impact of nutrients on chlorophyll concentrations. Along those lines, I’d suggest in Line 350-251 you specific that “No relationships between flow-adjusted inorganic nitrogen or TP loadings with chlorophyll-a were observed”

---

## Round 0.2 · accepted · Accept

After evaluating your responses to the 2 reviewers that suggested major revisions I am convinced that the paper is ready to go. Nice job considering and responding to the reviewers' comments!